# The Synergistic Effect of Lemongrass Essential Oil and Flometoquin, Flonicamid, and Sulfoxaflor on *Bemisia tabaci* (Genn.) (Hemiptera: Aleyrodidae): Insights into Toxicity, Biochemical Impact, and Molecular Docking

**DOI:** 10.3390/insects15050302

**Published:** 2024-04-24

**Authors:** Moataz A. M. Moustafa, Fatma S. Ahmed, Nawal Abdulaziz Alfuhaid, Nourhan A. El-Said, El-Desoky S. Ibrahim, Mona Awad

**Affiliations:** 1Department of Economic Entomology and Pesticides, Faculty of Agriculture, Cairo University, Giza 12613, Egypt; moat_mon@agr.cu.edu.eg (M.A.M.M.); fatma.sherif@cu.edu.eg (F.S.A.); nourhanadel158@gmail.com (N.A.E.-S.); aldosokyibrahim@yahoo.com (E.-D.S.I.); 2Department of Biology, College of Science and Humanities, Prince Sattam Bin Abdulaziz University, Al-Kharj 11942, Saudi Arabia; n.alfuhaid@psau.edu.sa

**Keywords:** whitefly, flometoquin, flonicamid, sulfoxaflor, lemongrass EO, molecular docking, IPM

## Abstract

**Simple Summary:**

As whiteflies, *Bemisia tabaci* (Genn.), are continually developing resistance to conventional insecticides, it is crucial and pressing to seek new approaches to management. In this study, the synergistic effect of lemongrass essential oil and flometoquin, flonicamid, and sulfoxaflor on *B. tabaci* was studied. Based on the found LC values, the decreasing order of toxicity to *B. tabaci* was as follows: sulfoxaflor > flonicamid > flometoquin > lemongrass EO. Sulfoxaflor and flonicamid exhibited significant inhibition of AchE activity, while only flonicamid demonstrated a significant impact on α-esterase. However, none of the tested compounds affected cytochrome P450 or GST. Additionally, lemongrass EO and the tested insecticides exhibited significant binding affinity to AchE.

**Abstract:**

The whitefly, *Bemisia tabaci* (Genn.), is one of the most dangerous polyphagous pests in the world. Eco-friendly compounds and new chemical insecticides have gained recognition for whitefly control. In this study, the toxicity and biochemical impact of flometoquin, flonicamid, and sulfoxaflor, alone or combined with lemongrass essential oil (EO), against *B. tabaci* was studied. In addition, a molecular docking study was conducted to assess the binding affinity of the tested compounds to AchE. Based on the LC values, the descending order of the toxicity of the tested compounds to *B. tabaci* adults was as follows: sulfoxaflor > flonicamid > flometoquin > lemongrass EO. The binary mixtures of each of the tested compounds with lemongrass EO exhibited synergism in all combinations, with observed mortalities ranging from 15.09 to 22.94% higher than expected for an additive effect. Sulfoxaflor and flonicamid, alone or in combination with lemongrass EO, significantly inhibited AchE activity while only flonicamid demonstrated a significant impact on α-esterase, and none of the tested compounds affected cytochrome P450 or GST. However, the specific activity of P450 was significantly inhibited by the lemongrass/sulfoxaflor mixture while α-esterase activity was significantly inhibited by the lemongrass/flometoquin mixture. Moreover, the lemongrass EO and all the tested insecticides exhibited significant binding affinity to AchE with energy scores ranging from −4.69 to −7.06 kcal/mol. The current findings provide a foundation for utilizing combinations of essential oils and insecticides in the integrated pest management (IPM) of *B. tabaci*.

## 1. Introduction

*Bemisia tabaci* Genn. (Hemiptera: Aleyrodidae) is an omnivorous insect pest that primarily targets Solanaceae, Cucurbitaceae, Cruciferae, leguminous vegetables, and some flower crops. With a host range of over 600 plant species, it causes crop damage by extracting plant juice, resulting in yellow spots, leaf yellowing, and defoliation, and it indirectly transmits more than 15 plant viruses, which cause 40 plant diseases [1,2,3,4,5]. According to the recommendations of the Agriculture Pesticide Committee (APC) of Egypt (https://agricultureegypt.com/uploads/Committees/PDF/Recomm2023.pdf) (accessed on 22 April 2024), six main types of chemical insecticides are commonly used for managing B. tabaci. These include (1) neonicotinoids such as thiamethoxam, imidacloprid, acetamiprid, and dinotefuran, (2) insect growth regulators like pyriproxyfen, (3) synthetic pyrethroids such as etofenprox, (4) ryanodine receptor modulators like cyantraniliprole, (5) biorational chemicals including azadirachtin, pymetrozine, spiromesifen, and orange oil, and (6) insecticide combinations such as abamectin + thiamethoxam and lambda-cyhalothrin + thiamethoxam. This categorization provides clarity regarding the variety of chemical insecticides commonly employed in the management of *B. tabaci* [6]. However, long-term use of these insecticides has been reported to trigger *B. tabaci* to acquire resistance to these chemicals, providing challenges in its prevention and management [7,8,9,10,11].

To effectively manage the escalating levels of resistance, it is crucial to develop integrated pest management (IPM) strategies. This involves a systematic rotation of various agents, encompassing both chemical agents, such as conventional insecticides, and non-chemical alternatives like essential oils, microbes, and entomopathogenic nematodes [12,13,14]. This method of rotating management agents has proven successful in decelerating the development of resistance [15]. Furthermore, as reported by [16], the use of selective insecticides with innovative modes of action, in conjunction with environmentally friendly chemicals such as essential oils (EOs), presents a viable approach for constructing long-term IPM strategies. Furthermore, EOs, which are biodegradable and often have lesser toxicity to non-target organisms, aid in ensuring the sustainability of such integrated techniques [17].

In the context of managing sap-feeding insects, which represent considerable challenges to crop production, the synergistic effects of combining chemical pesticides with EOs have yielded promising results [18,19]. However, assessing the risks of insecticide/EO combinations is critical, requiring a thorough examination of potential negative effects on non-target species, ecosystem dynamics, and human health. Recent studies [20,21] emphasized the significance of ongoing research to refine and optimize combinations, ensuring their efficacy and safety within the ever-changing landscape of sustainable pest management approaches.

Lemongrass EO has recently demonstrated a high level of toxicity to various pests. The insecticidal properties of lemongrass EO to black cutworm *Agrotis ipsilon*, cotton leafworm *Spodoptera littoralis,* velvet caterpillar *Anticarsia gemmatalis*, greater wax moth *Galleria mellonella*, and cowpea weevil *Callosobruchus maculatus* have been demonstrated [22]. Despite the promising results of various plant essential oils (EOs) against *B. tabaci* [11], the use of commercial biopesticides based on EOs is still limited, indicating a potential for their incorporation into contemporary agricultural practices [23].

Flometoquin, a novel insecticide distinguished by its common phenoxy-quinoline structure, was discovered in 2004 collaboratively by Nippon Kayaku and Mitsui Chemicals Crop & Life Solutions, Inc. Flometoquin acts as a mitochondrial complex III transport inhibitor—targeting the QI site [24] and preventing the utilization of energy by cells [25]. According to [26], flometoquin showed high effectiveness against the first-instar nymphs and adults of *B. tabaci*, *T. tabaci*, and *F. occidentalis*, the third-instar larvae of *P. xylostella*, and all developmental stages of *T. palmi*. Furthermore, the compound had no negative impacts on non-target arthropods.

As a selective insecticide, flonicamid is designed for the management of sap-sucking insects [27,28]. Its effect is caused by obstructing the feeding of insects. According to [29], this chemical belongs to the pyridine carboxamide group, which was discovered by Ishihara Sangyo Kaisha (ISK) Ltd. This compound is currently available globally under a variety of trade names, with markets in over forty countries across the Americas, Europe, Asia, and Africa. Furthermore, flonicamid effectively controls hemipterous insects, with notable success against Aphis gossypii Glover, as documented by [30], *Myzus persicae*, according to [31], and the plant hopper *Nilaparvata lugens*, as reported by [27].

Sulfoxaflor, the first compound in the developing sulfoximine insecticide category, works as a competitive modulator of nicotinic acetylcholine receptors (nAChRs) and is classified as a Group 4C insecticide by the Insecticides Resistance Action Committee (IRAC). This group is well-known for successfully managing a wide spectrum of sap-feeding insects [28,32]. In contrast to other Group 4 insecticides, sulfoxaflor has distinct interactions with nAChRs and metabolic enzymes, and these interactions cause major variations in the prevalence and severity of cross-resistance between sulfoxaflor and other insecticides.

In current research, we paired lemongrass EO with the three insecticides, flometoquin, flonicamid (as a novel chemical insecticide not commonly used for whitefly management), and sulfoxaflor (as a confirmed effective agent against whitefly and included for comparison). The objectives of this study were to (1) evaluate the toxicity of lemongrass EO and the three insecticides to adult whiteflies (*B. tabaci*); (2) assess the joint effects of lemongrass EO and the tested insecticides at sublethal concentrations; (3) investigate the impact of the tested chemicals, whether alone or as binary mixtures, at lethal and sublethal concentrations, on detoxification enzymes; and (4) conduct molecular docking analyses of the tested chemicals and the co-crystallized ligand ACT (acetate ion) with the active site of the AChE pocket.

## 2. Materials and Methods

### 2.1. Insect Population

A laboratory strain of whitefly, *Bemisia tabaci*, (Genn.), was reared on cotton seedlings (*Gossypium hirsutum* L.) in net-covered cages, under standard conditions of 26 ± 1 °C, 65 ± 5% relative humidity, and 16:8 L/D photoperiod, at the Economic Entomology and Pesticides Department, Faculty of Agriculture, Cairo University, Giza, Egypt. The cotton plants (cv. Giza 85) were grown in plastic pots containing a mixture of clay soil, peat moss, and sand at a ratio of 1:1:1. They were kept in an environmental chamber until the seedlings reached a height of 15–20 cm. This was done to keep the cotton seedlings away from any whitefly infestation.

### 2.2. Insecticides and Chemicals

In this study, three insecticides were used. Flometoquin (Kagura^®^ 10% SC) was supplied by Nippon Kayaku CO. Ltd., Tokyo, Japan, flonicamid (Teppeki^®^ 50% WG) was supplied by Anhui Sida Pesticide Chemical Co. Ltd., Bengbu, China, and sulfoxaflor (Closer^®^ 24% SC) was provided by Cortiva Agri-Science, Wilmington, DE, USA. The test chemicals (acetylthiocholine iodide (ATChI), 5,5-dithiobis (2-nitrobenzoic acid) (DTNB), α-naphthyl acetate, fast blue B salt, P-nitroanisole (PNOD), β-nicotinamide adenine dinucleotide phosphate (reduced β NADPH), L-glutathione reduced (GSH), and 1-chloro-2,4-dinitrobenzene (CDNB) were supplied by Sigma-Aldrich (Sigma-Aldrich, St. Louis, MO, USA).

### 2.3. Essential Oil Extraction and GC–MS Identification

Lemongrass essential oil, sourced from fresh *C. citratus* leaves, was extracted by hydrodistillation using a Clevenger-type apparatus, as described in the previously published papers of [33]. The chemical composition of the oil was identified using a GC Ultra-ISQ mass spectrometer (Thermo Scientific, Austin, TX, USA), and the results are presented in Appendix A.

### 2.4. Bioassays

To determine the LC values of the tested compounds to adult *B. tabaci*, the leaf-dip bioassay technique was used according to the modified method of IRAC number 015 (https://irac-online.org/content/uploads/Method_015_v3_june09.pdf) (accessed on 22 April 2024). In accordance with this method, discs (30 mm in diameter) from *Solanum lycopersicum* (tomato plants) were immersed in serial dilutions of insecticides with Tween-20 for 20 s. They were then air-dried for an hour and laid adaxially on a bed of 1% agar, which had been poured into the base of a Petri dish (30 mm in diameter, 20 mm high). The Petri dishes were ventilated through four holes covered with a metal screen. Six serial concentrations of lemongrass EO (ranging from 500 to 31.25 mg/L) and five serial concentrations of flometoquin (from 100 to 6.25 mg/L), flonicamid (from 25 to 1.56 mg/L), and sulfoxaflor (from 12 to 0.75 mg/L) were tested. These concentrations were diluted using water and the surfactant polysorbate Tween-20 (0.5%). Control discs were dipped only in distilled water for 20 s. Adult *B. tabaci* were removed from the rearing cage using a small mouth aspirator, and approximately 20 adults were placed in each Petri dish. The exact number of healthy adults in each Petri dish was recorded. The test insects were kept under the same environmental conditions as the untreated ones. With sufficient care, the natural mortality never exceeded 10% in the untreated check. The mortality data were assessed after 48 h of exposure, and insects were classified as dead if they showed no sign of movement.

### 2.5. Acute Toxicity

The acute toxicity of the tested insecticides, alone or as binary mixtures with the EO, to *B. tabaci* adults was assessed. The calculated LC_25_ of each tested compound was prepared and tested alone or in combination with the EO (LC_25_ of EO + LC_25_ of each compound). The same bioassay method described above was followed and mortality data were assessed 48 h post-treatment.

### 2.6. Biochemical Assays

#### 2.6.1. Preparation of *B. tabaci* Homogenate

Adults of *B. tabaci* were exposed to the median lethal concentrations (LC_50_) or sub-lethal concentrations (LC_25_) of the tested compounds individually or using the binary mixtures with lemongrass EO (LC_25_:LC_25_). After 48 h, live adults were collected to determine the activity of detoxification enzymes. Activities of acetylcholinesterase (AchE), cytochrome P450 (P450), α-esterase, and glutathione S-transferase (GST) were determined in treatments and control. Three replicates for each treatment were used. Each replicate (0.03 g adults) was homogenized in a cold 0.3 mL homogenization buffer (0.1 M phosphate buffer, pH 7.0). The homogenates were then centrifuged at 12,000 rpm for 15 min, and the supernatants were transferred into a clean Eppendorf tube [6,23,26]. The protein content of each replicate was estimated using Bradford’s method [34].

#### 2.6.2. Acetylcholine Esterase (AChE) Assay

The AChE activity was analyzed as described by [35]. A hundred microliters of the enzyme source and 50 µL of 0.075 M acetylthiocholine iodide (ATChI) were used as the testing substrate. The reaction was initiated by adding 50 µL of 0.01 M dithio-bis-nitro benzoic acid (DTNB). Measurements were taken at intervals of 1 minute for 5 min at a wavelength of 412 nm.

#### 2.6.3. Cytochrome P450 Assay

The P450 assay was conducted using the method described by [36]. The reaction mixture consisted of 100 µL of 2 mM p-nitroanisole and 90 µL of the enzyme solution at 27 °C for 2 min. Then, 10 µL of 9.6 mM NADPH was added to initiate the reaction. The absorbance was recorded at a wavelength of 405 nm, against the p-nitrophenol standard curve.

#### 2.6.4. Esterase Assay

Determination of α- esterase activity was carried out as reported by [37]. The reaction mixture consisted of 30 µL of the enzyme source, 820 µL of 40 mM potassium phosphate buffer (pH 7), and 30 mM of α-naphthyl acetate (α-NA) as the testing substrate. The mixture was incubated for 15 min at 27 °C. After incubation, 2% fast blue B was added to stop the reaction. The absorbance was measured at 600 nm, using α-naphthol as a reference standard.

#### 2.6.5. Glutathione S-transferase (GST) Assay

The glutathione S-transferase (GST) assay was quantified according to [38]. The reaction solution was prepared with 10 µL of the enzyme source, 25 µL of 30 mM 1-chloro-2,4-dinitrobenzene (CDNB) as the substrate, and 25 µL of 50 mM glutathione, and supplemented to 1 mL with 50 mM potassium phosphate buffer (pH 6.5). The optical density was monitored at a wavelength of 340 nm, with readings taken at one-minute intervals for a total duration of five minutes. The GST activity was calculated using the extinction coefficient (ε, 9.6 mM^−1^ cm^−1^) [39].

### 2.7. Molecular Docking Study

Molecular docking for the proposed compounds was performed against the active site of AChE (PDB ID:6XYS). The crystal structure of the acetylcholine esterase (AChE) enzyme was downloaded from the protein data bank (http://www.rcsb.org) (accessed on 22 April 2024). Gaussian 09 software was used to generate a file containing the structures of the tested compounds in the PDB format. The molecular docking studies were carried out using the MOE software (2015). The co-crystallized ligand was re-docked in its original enzyme structure using the default parameters. 

### 2.8. Data Analysis

SPSS (V.22) was utilized in data analysis. Data were coded, entered, and examined for satisfying the parametric tests’ assumptions. Shapiro–Wilk and Kolmogorov–Smirnov tests for normality were used with continuous variables. Data are presented as mean ± SD. ANOVA was performed for the control and the experiments. Regarding enzymatic activity, at least three replicates were analyzed for each group. The interaction between lemongrass EO and the tested insecticides on *B. tabaci* adults was determined using the χ^2^ test [40] using the following formula:$$\chi^{2}=\sum \frac{{\left(O-E\right)}^{2}}{E}$$
where χ^2^ is the chi-square test statistic, *O* is the observed mortality, and *E* is the expected mortality. The formula used to calculate the expected additive proportional mortality (ME) for EO/insecticide combinations is
M_E_ = M_N1_ + M_N2_ (1 − M_N1_)
where M_N1_ and M_N2_ represent the observed proportional mortalities caused by the EO and insecticide alone, respectively. The results from the χ^2^ test, where χ^2^ = (M_N1_ + 2 − ME)^2^/ME and MNC is the observed mortality for the EO/insecticide combination, were compared to the tabulated χ^2^ value for 1 degree of freedom. If the calculated χ^2^ value exceeded the tabulated value, a non-additive effect between the two combined compounds was suspected [40]. If the difference M_N1_ + 2 − ME was positive or negative, the interaction was considered synergistic or antagonistic, respectively. Conversely, if the tabulated χ^2^ value exceeded the calculated one, an additive effect was considered (at *p* < 0.05).

Data from enzyme assays of each compound at the LC_50_ values were subjected to ANOVA and then to Dunnett’s pairwise comparison test between the control and each tested compound. Meanwhile, data from enzyme assays of the binary mixtures were subjected to Tukey’s multiple comparison tests between all tested groups. A *p*-value of 0.05 was considered statistically significant. When needed, data were visualized with GraphPad Software Inc., San Diego, CA, USA.

## 3. Results

### 3.1. Toxicity of the Tested Compounds to Bemisia tabaci

The leaf-dip bioassay technique was utilized to examine the toxicity of lemongrass, flometoquin, flonicamid, and sulfoxaflor to *B. tabaci* adults. As shown in Table 1, the LC_25_ and LC_50_ values of lemongrass, flometoquin, flonicamid, and sulfoxaflor were 68.85 and 147.71, 11.79 and 22.57, 3.98 and 7.52, and 1.89 and 3.69 mg/L, respectively.

### 3.2. Interaction of Lemongrass EO with the Tested Insecticides on B. tabaci Adults

Data in Table 2 show the joint toxicity of lemongrass EO (LC_25_) and each of the tested insecticides (LC_25_) to *B. tabaci* adults. The interaction showed synergistic effects in all combinations, resulting in observed mortalities higher than expected for an additive effect, ranging from 15.09% to 22.94%.

### 3.3. Biochemical Impact of the Tested Compounds

As shown in Figure 1, Both flonicamid and sulfoxaflor exhibited significant inhibition of AchE-specific activity (F (4,10) = 4.48, *p* < 0.05) while only flonicamid demonstrated a significant impact on α-esterase-specific activity (F (4,10) = 4.84, *p* < 0.05). On the other hand, neither cytochrome P450- nor GST-specific activity was significantly affected by any of the tested compounds, when compared to the control group (Figure 1 and ANOVA Appendix A).

### 3.4. Biochemical Effect of Binary Combinations of Lemongrass EO with the Tested Insecticides

As shown in Figure 2, AchE-specific activity was significantly inhibited by the LC_25_ of flometoquin, flonicamid, and sulfoxaflor, when administered individually, as well as by the binary mixtures of lemongrass/flometoquin (F (3,8) = 7.39, *p* ≤ 5) and lemongrass/sulfoxaflor (F (3,8) = 6.48, *p* ≤ 5). P450-specific activity was significantly inhibited by the LC_25_ of flonicamid alone (F (3,8) = 8.23, *p* ≤ 5), and by the binary mixture of lemongrass/sulfoxaflor (F (3,8) = 4.74, *p* ≤ 5) (see ANOVA Appendix A).

α-esterase-specific activity was significantly inhibited only by the binary mixture of lemongrass/flometoquin (F (3,8) = 6.45, *p* ≤ 5). On the other hand, there was no significant observed difference in GST-specific activity between either the individual insecticides or their binary mixtures with lemongrass EO, compared to the control group (Figure 2).

### 3.5. Docking Mechanisms of the Tested Compounds with AchE Enzyme

The docking process of the tested compounds was validated by running the docking procedure for the co-crystallized ligand ACT (acetate ion) against the active site of the pocket. The tested compounds gave good energy scores ranging from (S) = −4.69 to −7.06 kcal/mol (Table 3). The proposed binding pattern of citral, the major bioactive component of lemongrass EO, revealed one hydrogen bond with the amino acid residue TYR 324 and two H–arene contacts with TRP 321 (Figure 3). The interaction between flometoquin and AChE enzyme was stabilized through one hydrogen bond with TYR 73, an arene–H interaction with the residue TYR 324, and pi–pi stacking with TYR 71 amino acid (Figure 3). Flonicamid combined with the receptor through one hydrogen bond with LEU 328 and an arene–H contact with TYR 324 amino acid (Figure 3). Sulfoxaflor showed one hydrogen bond with PHE 371 amino acid and H–arene interaction with the residue TYR 324 (Figure 3). ACT had an energy score (S) = −3.41 kcal/mol and produced two H-bonds with PHE 371 and TYR 374 amino acids (Figure 3).

## 4. Discussion

Crop production has been accompanied by the widespread use of various pesticides for maintaining crop productivity while minimizing pest damage [41]. However, the frequent increase in pesticide applications has not resulted in large decreases in pest populations; rather, it has contributed to the development of pest resistance to routinely used pesticides [42,43]. When managing a particular pest, the fundamental purpose of pest management combinations is to minimize the concentration of each component while maintaining a high level of efficacy. This is always compared to using each element separately at the same concentration. The discrepancy in the metabolic pathways activated within the treated organism between individual application of any compound and its mixture with another agent represents a significant limitation to the use of pesticide mixtures in pest management programs. Nevertheless, chemical mixtures, particularly those which combine EOs with low doses of chemical insecticides, are viewed as a promising strategy for insect resistance management (IRM) [12]. There are several studies that have evaluated mixtures of EOs and insecticides [44,45,46,47,48,49]. However, the current study represents the initial exploration of the impact of lemongrass EO combined with different pesticides on the metabolic enzymes in *Bemisia tabaci* adults.

Interestingly, all the combinations tested here showed synergistic advantages. The most synergistic combination was lemongrass/sulfoxaflor, followed by lemongrass/flonicamid and lemongrass/flometoquin. These findings indicate that combining lemongrass EO with an insecticide might potentially restore the insecticide’s potency against *B. tabaci*. This finding is consistent with that of [50], who suggested combining neonicotinoids with buprofezin or pyriproxyfen at a ratio of 1:1 to restore their potency against *B. tabaci*. However, these combinations must be tested on nontarget organisms, such as honeybees, to evaluate their safety.

Furthermore, the potency of EOs against the notorious *B. tabaci* has been the focus of several researchers. For example, ref. [11] shed light on the potent effects of three distinct mustard EOs which demonstrated LC_50_ values of 0.73%, 1.02%, and 1.05% (equivalent to 7300, 10,020, and 10,050 mg/L) on *B. tabaci* eggs, indicating their potential as a management agent against *B. tabaci*. The same oils showed efficacy, with LC_50_ values of 0.69%, 0.65%, and 0.88% (equivalent to 6900, 6500, and 8800 mg/L) on young nymphs and 1.03%, 0.91%, and 0.90% (equivalent to 10,300, 9100, and 9000 mg/L) on old nymphs, respectively. Ref. [51] studied the EOs of *Piper marginatum* and *Mansoa alliacea* and reported LC_50_ values of 9.39 μL mL^−1^ and 10.99 μL mL^−1^ (equivalent to 9390–10,990 mg/L) to *B. tabaci* nymphs, thus demonstrating their effectiveness against this pest. Ref. [52] also contributes to the discussion, revealing that *Zanthoxylum riedelianum* EO, in the range of 1000–20,000 mgL^−1^ (0.1–2%), resulted in a significant reduction in white fly eggs by 53.2–98.3%, highlighting its pest control potential. All these valuable studies demonstrated the possibility of integrating EOs into management programs of *B. tabaci*. However, none of these EOs showed a LC_50_ value of less than 1000 mg/L. On the contrary, lemongrass EO, in the current study, exhibited extraordinary toxicity against *B. tabaci* adults, with a remarkable LC_50_ value of 147.71 mg/L. This promising result encourages the future incorporation of this environmentally friendly compound into comprehensive pest management programs, heralding a potential breakthrough in sustainable pest control strategies.

In the current study, the LC_50_ value of flometoquin against Bemisia tabaci adults was determined to be 11.79 mg/L. However, it was determined to be 0.79 mg/L, a significantly lower value, by [26]. While there is a discrepancy between the LC_50_ values in this study and the referenced one, it is important to note that different *B. tabaci* populations may react differently to flometoquin. Flometoquin is a new compound, and the findings of [26] contribute to the ongoing investigation into its effectiveness, particularly considering its safety for non-target arthropods and its efficacy against various pest species.

In the current study, flonicamid showed a highly toxic effect on *B. tabaci* adults, with an LC_50_ of 7.52 mgL^−1^. Ref. [53] found that flonicamid LC_50_ ranged from 95.7 to 1001 mg L^−1^ to whitefly field populations. Notably, the laboratory strain had a significantly lower LC_50_ (784 mgL^−1^) than some field-collected strains. The lower LC values observed in our study may indicate the high efficacy of flonicamid against *B. tabaci* in Egypt, which supports the inclusion of this compound in the management program of this pest.

Additionally, in this research, sulfoxaflor exhibited the highest toxicity to *B. tabaci* adults with an LC_50_ of 3.69 mg/L Ref. [54] noted that sulfoxaflor has demonstrated effectiveness against various sap-feeding insects, including those resistant to neonicotinoids and other insecticides. However, due to safety concerns regarding the Egyptian honeybee *Apis mellifera* at certain field concentrations of sulfoxaflor [55,56], it might not be advisable to solely rely on this compound despite its potent toxicity. Nonetheless, employing lower concentrations of the compound could potentially address this issue.

Regarding the biochemical effect of the tested compounds, lemongrass EO alone did not affect any of the detoxification enzymes while its binary mixtures with the tested insecticides significantly affected some of them. Flometoquin and sulfoxaflor, alone or as a mixture with lemongrass EO, inhibited the AchE activity and similarly flometoquin inhibited α-esterase activity. Thus, the inhibitory effect of these mixtures cannot be attributed to the joint effect. On the other hand, sulfoxaflor alone did not affect P450-specific activity while its mixture with lemongrass EO showed significant inhibition. Thus, the high mortality caused by this mixture (Table 3) is more likely attributed to this inhibitory effect on detoxification enzymes in general.

In the present study, while the lemongrass/flonicamid combination showed a strong synergistic effect, it did not exhibit any notable impact on detoxification enzymes. This suggests that the concentrations or proportions of the EO and insecticide in the mixture could affect how they interact with detoxification enzymes. Additionally, no significant effects were observed on GST-specific activity across all treatments, which suggests that the tested compounds, alone or as mixtures, do not influence phase 2 metabolizing enzymes. This inconsistency might be due to variations in the metabolic pathways or enzyme systems associated with detoxification mechanisms between *B. tabaci* adults and other insects [57].

Acetylcholinesterase (AchE) is a key enzyme that breaks down the acetylcholine into choline and acetate and, hence, is a relevant biomarker in the identification of neuromuscular toxic effects of the pesticides targeting this enzyme [58,59]. The present study revealed that sulfoxaflor at its lethal concentration (LC_50_) and flonicamid at its sub-lethal and lethal concentrations (LC_25_ and LC_50_) significantly inhibited the AchE-specific activity of *B. tabaci* adults 48 h post-treatment. Sulfoxaflor, like neonicotinoids, impairs the neurotransmitter acetylcholine-related neuronal functions by its agonist effects. Thus, it is reasonable that the increased and continuous agonist conjugation of sulfoxaflor inhibited AchE-specific activity. In addition, recent studies reported that neonicotinoids resulted in inhibition of AchE activity in German cockroach females [60], fish [61], and mammals [62]. In contrast, sub-lethal doses of either sulfoxaflor or neonicotinoids are also reported to increase AChE activity in bees [55], zebrafish [63], and other invertebrates exposed to it [64]. The most surprising result in the current study was the inhibition of AchE in *B. tabaci* adults 48 h after treatment with both lethal and sub-lethal concentrations of flonicamid. Notably, there were no observed poisoning symptoms such as convulsions or knockdown after treatment. In addition, refs. [29] also reported that flonicamid exhibited no effect against acetylcholinesterase. It is noteworthy that the Insecticide Resistance Action Committee (IRAC) (https://irac-online.org/) (accessed on 22 April 2024) initially classified flonicamid as IRAC Group 9 along with pymetrozine; both of these insecticides have been identified as chordotonal organ influencers [65]. While pymetrozine’s target was later identified as chordotonal organ TRPV channels [66], flonicamid, unlike pymetrozine, did not activate insect TRPV channels or compete with TRPV activators for binding, as detailed by [67]. As a result, flonicamid was reclassified as part of IRAC Group 29 whose insecticides have an undefined target site of action. These intriguing findings strongly indicate that flonicamid has a unique molecular target, and this may increase the possibility of varied effect of flonicamid in different species.

In the current work, to validate the relationship between AchE enzyme inhibition and the tested insecticide, we conducted a molecular docking study on the tested compounds against the active site of the acetylcholinesterase (AChE) enzyme and acetate ion (ACT). The aim was to understand how the tested insecticides interact with the key amino acids of the AchE enzyme and their binding modes. Interestingly, flometoquin, despite its lower acute toxicity compared to sulfoxaflor and flonicamid, demonstrated a higher binding affinity to AChE. This was evidenced by an energy score of −7.06 Kcal/mol, in contrast to −5.60 and −5.26 Kcal/mol for the two tested insecticides, respectively. Moreover, these compounds, along with citral, the major component of lemongrass EO, exhibited a significant binding affinity to AChE (4.69 Kcal/mol), even higher than that of the ACT ligand.

Joining molecular docking results with those of biochemical assay signifies the importance of correlating the binding affinity of an insecticide to the target protein and the inhibition of this protein. The results of molecular docking are in line with those of the biochemical assays of the three tested insecticides and lemongrass EO. As illustrated in Figure 2, flometoquin, followed by sulfoxaflor and then flonicamid, significantly inhibited the AChE-specific activity, compared to the control group. Conversely, the inhibition caused by lemongrass EO was non-significant.

Furthermore, flometoquin, according to the IRAC, is categorized as a mitochondrial complex III electron transport inhibitor, while flonicamid is identified as a chordotonal organ nicotinamidase inhibitor. Despite their disparate target sites relative to sulfoxaflor, which is deemed a nicotinic acetylcholine receptor (nAChR) competitive modulator, they exhibited high binding affinity to AChE. The elevated binding affinity of flometoquin and flonicamid to AChE suggests that AChE might play a critical role in the toxic action of these compounds on *B. tabaci*. To the authors’ knowledge, this is the first molecular docking study of flometoquin and flonicamid insecticides against the active site of the AChE enzyme. However, further molecular studies are necessary to elucidate the relationship between the toxic actions of flometoquin and flonicamid and their potential inhibitory effects on AChE.

## 5. Conclusions

In general, the use of lemongrass essential oil as a biological control agent in combination with sulfoxaflor, flonicamid, or flometoquin showed a synergistic effect on adult whiteflies (*B. tabaci*). This means that the combined effect of the EO and the insecticides was greater than the sum of their individual effects. However, these findings indicate a need for further research. More in-depth molecular studies could help better understand these results by exploring how the EO and insecticides interact with detoxifying enzymes in different concentrations or ratios. Furthermore, a detailed examination of how adult whiteflies detoxify these substances could provide valuable insights.

## Figures and Tables

**Figure 1 insects-15-00302-f001:**
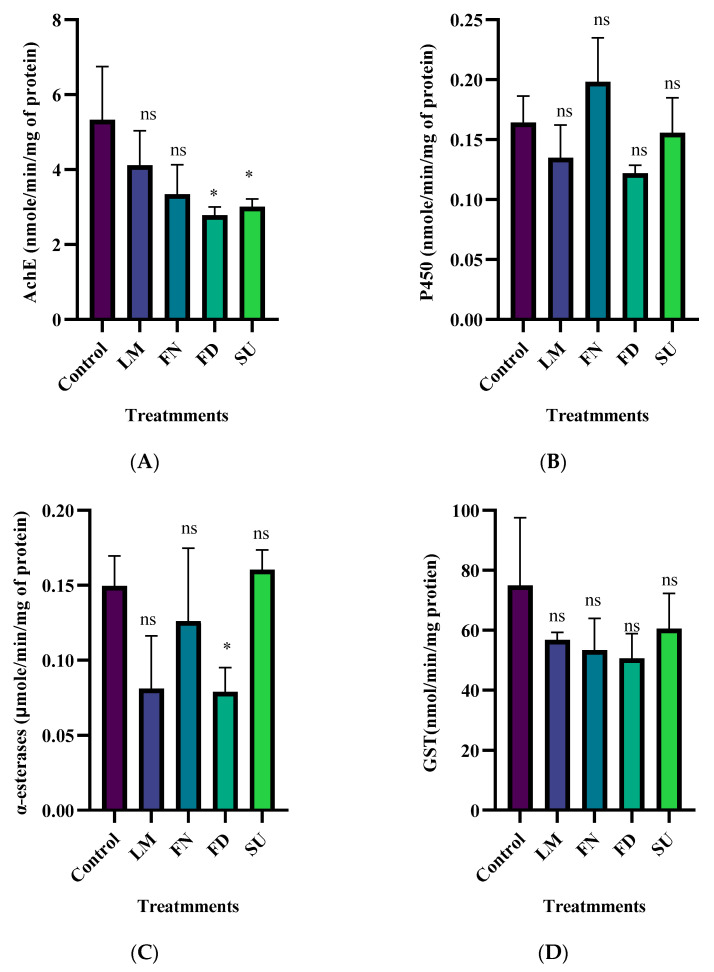
Specific activity of (**A**) AchE, (**B**) P450, (**C**) α-esterase, and (**D**) GST in *Bemisia tabaci* adults 48 h after treatment with lemongrass essential oil (EO), flometoquin (FN), flonicamid (FD), and sulfoxaflor (SU) at their LC_50_ values, compared with the control group. Dunnett’s multiple comparison test was performed, with the data expressed as mean ± S.D. * Indicates *p*  <  0.05, ns: non-significant).

**Figure 2 insects-15-00302-f002:**
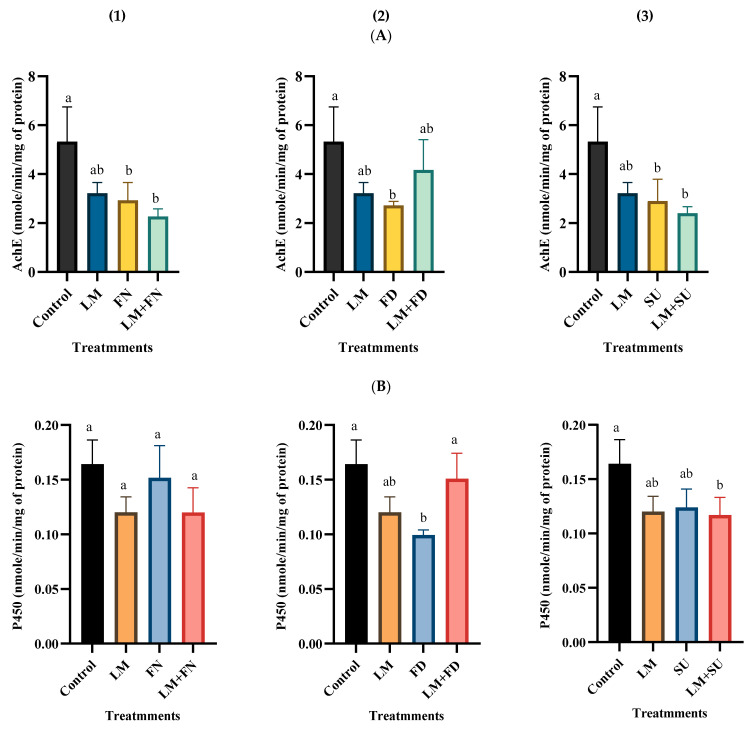
The specific activity of (**A**) AchE, (**B**) P450, (**C**) α-esterase, and (**D**) GST in *Bemisia tabaci* adults 48 h after treatment with individual insecticides ((1) flometoquin, (2) flonicamid, or (3) sulfoxaflor) or the insecticides combined with lemongrass essential oil, at their LC_25_ values, compared to the untreated control. Tukey’s multiple comparison test was performed with the data expressed as mean ± S.D. Mean values for the same index (bars of each graph) with unlike letters are significantly different (*p* < 0.05).

**Figure 3 insects-15-00302-f003:**
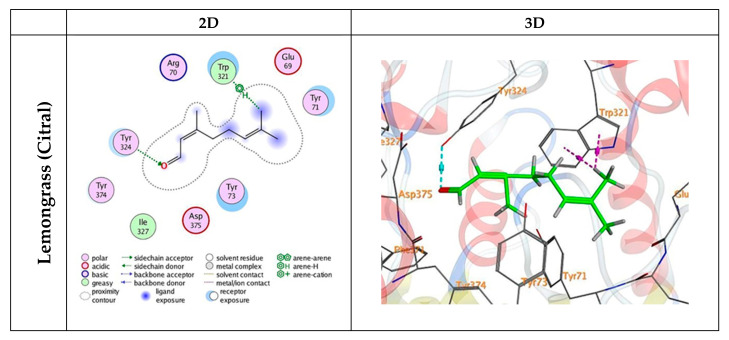
2D and 3D interactions of citral (the major bioactive component of lemongrass EO), flometoquin, flonicamid, sulfoxaflor, and acetate ion (ACT) in the active site of AChE enzyme (PDB ID:6XYS). Hydrogen bonds are displayed in cyan, and H–pi bonds are displayed in magenta.

**Table 1 insects-15-00302-t001:** Toxicity of lemongrass (*C. citratus*) EO, flometoquin, flonicamid, and sulfoxaflor insecticides to the white fly (*Bemisia tabaci)* adults 48 h post-treatment.

Treatments	LC_25_ (mg/L) (95% Confidence Limits)	LC_50_ (mg/L) (95% Confidence Limits)	Slope ± SE	χ^2^
Lemongrass (*C. citratus*)	68.85 (56.40–81.08)	147.71 (128.19–171.02)	2.03 ± 0.16	0.73
Flometoquin	11.79 (9.81–13.60)	22.57 (19.88–25.55)	2.37 ± 0.17	1.07
Flonicamid	3.98 (3.38–4.58)	7.52 (6.64–8.54)	2.44 ± 0.18	1.76
Sulfoxaflor	1.89 (1.60–2.19)	3.69 (3.25–4.22)	2.33 ± 0.18	1.94

**Table 2 insects-15-00302-t002:** Mortality (% ± SD) of the white fly (*Bemisia tabaci)* adults 48 h post-treatment with binary combinations of lemongrass (LM) essential oil and flometoquin (FN), flonicamid (FD), or sulfoxaflor (SU) at their LC_25_ values.

Compounds	Conc.	No.	% M ^a^	ME	χ^2 b^	D ^b^
LM	LC_25_	62	25.2 ± 5.5 ^b^			
FN	LC_25_	85	29.8 ± 9.2 ^b^			
FD	LC_25_	69	28.9 ± 10.7 ^b^			
SU	LC_25_	64	35.4 ± 5.0 ^b^			
LM + FN	LC_25_ + LC_25_	61	62.6 ± 3.5 ^a^	47.50	4.97	15.09
LM + FD	LC_25_ + LC_25_	71	66.7 ± 3.8 ^a^	46.86	8.64	19.88
LM + SU	LC_25_ + LC_25_	74	71.5 ± 4.3 ^a^	48.52	11.1	22.94

^a^ Means followed by the same letter within columns are not significantly different (*p* < 0.05). ^b^ D = difference between observed mortality and expected mortality for an additive effect: D significantly greater than 0 = synergistic interaction; D significantly smaller than 0 = antagonistic interaction.

**Table 3 insects-15-00302-t003:** Docking interaction data calculations of citral (the major bioactive component of lemongrass EO), flometoquin, flonicamid, sulfoxaflor, and ACT (acetate ion) inside AChE enzyme (PDB ID: 6XYS) active spots.

Compound	Energy Score (S) (Kcal/mol)	Affinity Bond Strength (Kcal/mol)	Affinity Bond Length (in A° from Main Residue)	Amino Acids	Ligand	Interaction
Citral	−4.69	−0.8	2.82	TYR 324	O 19	H-acceptor
		−0.6	3.79	TRP 321	C 20	H-pi
		−0.6	4.44	TRP 321	C 20	H-pi
Flometoquin	−7.06	−0.8	2.85	TYR 73	O 3	H-acceptor
		−0.6	4.00	TYR 324	6-ring	pi-H
		−0.0	3.97	TYR 71	6-ring	pi-pi
Flonicamid	−5.26	−0.8	3.44	LEU 328	N 22	H-acceptor
		−0.7	3.64	TYR 324	6-ring	pi-H
Sulfoxaflor	−5.60	−0.7	3.70	PHE 371	N 3	H-acceptor
		−0.6	4.53	TYR 324	C 14	H-pi
ACT (acetate ion)	−3.41	−1.8	3.00	PHE 371	O 6	H-acceptor
		−0.7	2.97	TYR 374	O 7	H-acceptor

## Data Availability

The datasets generated during and/or analyzed during the current study are available from the corresponding author upon reasonable request.

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
