# Peer review of "The Synergistic Effect of Lemongrass Essential Oil and Flometoquin, Flonicamid, and Sulfoxaflor on *Bemisia tabaci* (Genn.) (Hemiptera: Aleyrodidae): Insights into Toxicity, Biochemical Impact, and Molecular Docking"

_insects, 2024, doi:10.3390/insects15050302_

Round 1

Reviewer 1 Report

Comments and Suggestions for Authors

1. A laboratory strain was used in this study, It would be better if you can use field population to do.

2. The production cost of lemongrass essential oil may be higher than that of traditional chemical pesticides, and integrating lemongrass essential oil into pest management programs may bring additional expenses. However, this study did not delve deeply into the economic feasibility of this strategy, as well as its actual benefits and burdens for farmers, which may affect its application in large-scale agricultural production.

3. Although lemongrass essential oil is considered environmentally friendly, its potential impacts on non-target organisms still need further study, especially its long-term effects on the ecosystem level.

4. The image clarity is not sufficient, like figure 3. Please replace these figs,

5. what is the difference between lemongrass essential oil, lemongrass EO or lemongrass.

Comments on the Quality of English Language

Quality of English Language is OK to read and understand.

Author Response

The synergistic effect of lemongrass essential oil on flometoquin, flonicamid, and sulfoxaflor on Bemisia tabaci Genn. (Hemiptera: Aleyrodidae): Insights into toxicity, biochemical impact, and molecular docking

Authors' Responses to “Reviewer 1” comments

Dear Professor,

We extend our gratitude for dedicating your time to reviewing our manuscript and providing insightful feedback for its enhancement. We greatly value your constructive comments, which we have meticulously incorporated into our revised version. Below, we provide detailed responses to each of your suggestions.

Thank you once again for your invaluable contribution to the refinement of our work.

Warm regards,

Authors

  1. A laboratory strain was used in this study, It would be better if you can use field population to do.

****Authors Response

Agree!

Indeed, employing field populations may offer a more realistic depiction of the efficacy of tested chemicals, whether applied individually or in binary mixtures. However, our current manuscript focuses on assessing the toxic impact of a binary mixture consisting of lemongrass essential oil with flometoquin, flonicamid, or sulfoxaflor insecticides on Bemisia tabaci. To achieve this, we initially collected a field population of B. tabaci, which we subsequently maintained under controlled laboratory conditions for three consecutive generations. It's important to note that during this period, the insects were not exposed to the insecticides, rendering them a laboratory population rather than a susceptible one. The purpose behind maintaining B. tabaci in the laboratory was to ensure a uniform response to our tested materials while eliminating any possibility of prior exposure to the insecticides before collection. By doing so, we aimed to establish a baseline for comparison and to minimize confounding factors in our experimental setup.

Moreover, having confirmed the toxic effects of the lemongrass essential oil mixture with flometoquin, flonicamid, and sulfoxaflor in our current manuscript, our next step involves evaluating these mixtures under field conditions. This field trial aims to assess the practicality and effectiveness of these blends in real-world scenarios. We anticipate that the results of this field study will provide valuable insights into the applicability and performance of these mixtures outside the laboratory setting. Moving forward, we intend to document and submit the outcomes of our field experiments as a separate manuscript, thereby expanding on the findings presented in our current work.

  1. The production cost of lemongrass essential oil may be higher than that of traditional chemical pesticides, and integrating lemongrass essential oil into pest mangement programs may bring additional expenses. However, this study did not delve deeply into the economic feasibility of this strategy, as well as its actual benefits and burdens for farmers, which may affect its application in large-scale agricultural production.

****Authors Response

The recommended dose of the tested insecticides and lemongrass is as following;

For Flonicamid, Flometoquin, Sulfoxaflor and lemongrass is 59.5, 47.6, 23.8 and 476.2 g a.i./h respectively, while the production cost of these compounds is 12, 30, 10 and 12 US dollar /h. So, these calculation showed that the cost of using the lemongrass with sulfoxaflor are cheaper and less than the other tested insecticides and could be recommended to be used easily in the pest management on large scale.

  1. Although lemongrass essential oil is considered environmentally friendly, its potential impacts on non-target organisms still need further study, especially its long-term effects on the ecosystem level.

****Authors Response

Even though lemongrass essential oil hasn’t been tested on all non-target organisms, its inflammatory, antioxidant, antibacterial, and immunomodulatory effects have been evaluated in rats. In the study by Shalaby et al. (2023), lemongrass essential oil was utilized as a protective compound against the toxic impact of Perfluorooctane sulfonate—a chemical widely used in various industrial and commercial applications such as firefighting foams, stain-resistant coatings for fabrics, electronics manufacturing, and metal plating processes—on rats. The study suggested that lemongrass essential oil significantly mitigated the harmful effects of Perfluorooctane sulfonate by reducing oxidative, inflammatory, and apoptotic impacts.

Furthermore, in the study conducted by Júnior et al. ( 2024), the antioxidant and hypoglycemic effects of lemongrass essential oil were confirmed in a type 1 diabetes (DM1) rat model. They concluded that lemongrass essential oil shows promise as a therapeutic option for managing DM1.

  1. The image clarity is not sufficient, like figure 3. Please replace these figs,

****Authors Response

Thank you for bringing to our attention your concerns regarding the clarity of Figure 3. We understand the importance of high-quality visual representations to support the textual narrative and ensure the scientific integrity of our findings.

We have taken immediate steps to address this issue by enhancing the resolution of the figure in question to exceed 600 PPI (pixels per inch) as it was set at 300 PPI, which is well above the standard resolution for high-quality printed material. This adjustment will ensure that all the intricate details of the molecular interactions are crisply represented and easily discernible.

Additionally, we have reviewed all other figures in the manuscript to ascertain that each meets the high-resolution criteria, guaranteeing that our visual data is presented with the utmost clarity. The revised figures will facilitate better visualization of the binding sites and interactions described in our study, thus improving the reader's ability to follow the scientific argument laid out in our research.

  1. what is the difference between lemongrass essential oil, lemongrass EO or lemongrass.

****Authors Response

They're all the same! We abbreviated 'essential oil' to 'EO' to decrease its prominence in the manuscript.

References

Júnior, A.S.S., Aidar, F.J., Silva, L.A.S., de B. Silva, T., de Almeida, S.F.M., Teles, D.C.S., de L. Junior, W., Schimieguel, D.M., de Souza, D.A., Nascimento, A.C.S., Camargo, E.A., dos Santos, J.L., de O. e Silva, A.M., de S. Nunes, R., Borges, L.P., Lira, A.A.M., 2024. Influence of Lemongrass Essential Oil (Cymbopogon flexuosus) Supplementation on Diabetes in Rat Model. Life 14. https://doi.org/10.3390/life14030336

Shalaby, A.M., Albakkosh, A.M., Shalaby, R.H., Alabiad, M.A., Elshamy, A.M., Alorini, M., Jaber, F.A., Tawfeek, S.E., 2023. Lemongrass Essential Oil Attenuates Perfluorooctane Sulfonate-Induced Jejunal Mucosal Injury in Rat: A Histological, Immunohistochemical, and Biochemical  Study. Microsc. Microanal. Off. J. Microsc. Soc. Am. Microbeam Anal. Soc. Microsc. Soc. Can. 29, 841–857. https://doi.org/10.1093/micmic/ozad009

Reviewer 2 Report

Comments and Suggestions for Authors

The manuscript “The synergistic effect of lemongrass essential oil on flometoquin, flonicamid, and sulfoxaflor on Bemisia tabaci Genn. (Hemiptera: Aleyrodidae): Insights into toxicity, biochemical impact, and molecular docking” (insects-2967128) reported the synergistic effect of lemongrass essential oil on flometoquin, flonicamid, and sulfoxaflor on B. tabaci. The results of this study provide valuable information for mitigating the current status of insecticide resistance in B. tabaci. Here are some issues for consideration before recommending for publishing.

1. The authors failed to provide a coherent rationale for the selection of lemongrass essential oil as a promising candidate in the Introduction section. It is recommended that the investigation into the utilization of lemongrass essential oil for pest control and managing insecticide resistance be incorporated.

2. The mode of action and target site of Flometoquin, a newly developed insecticide, necessitate detailed elucidation within scholarly discourse.

3. In the Materials and Methods section, it is imperative for the authors to furnish a comprehensive account of the protocol utilized to ascertain detoxification enzyme activity, encompassing key variables like incubation duration, temperature conditions, and pH levels. Such meticulous documentation serves to bolster the integrity of the findings, notably in substantiating the absence of noteworthy discrepancies in GST and P450 enzyme functionalities. The following reference might be helpful: Journal of Agricultural and Food Chemistry, 2023, 71, 5230−5239.

4. Please confirm the presentation of Figure 1, ensuring uniformity in the measurement units of enzyme activities across both Figure 1 and Figure 2.

5. The reliability of the control group within the results of GST enzyme activity appears to be insufficient; it is advisable to carefully review the provided data.

6. Based on the discoveries outlined in sections 3.3 and 3.4, it is imperative to elucidate and deliberate on the diverse patterns noted in enzyme activities subsequent to insecticide treatments at LC25 and LC50 levels. Notably, the marked reduction in AChE activity post-LC25 flometoquin exposure, juxtaposed with the absence of notable variance in the context of LC50 treatment, demands additional scrutiny and discourse. Analogous patterns evident in alternative treatments similarly necessitate thorough contemplation.

7. The study employed the active sites of AChE in Drosophila melanogaster for analysis. Consideration is suggested for utilizing those in B. tabaci or conducting a comparative evaluation of the active sites between the two species. A more detailed account of the statistical analysis of docking results is recommended within the Material and Methods section.

8. While the scoring of proteins and small molecules typically relies on the Affinity Bond strength parameter, the authors' criterion of the Energy score reveals diverging trends compared to the Affinity Bond strength.

9. The visual representation depicted in Figure 3 exhibits substandard quality as the binding sites remain imperceptible. An in-depth analysis centering on the binding site is imperative to enhance the scholarly discourse.

Author Response

The synergistic effect of lemongrass essential oil on flometoquin, flonicamid, and sulfoxaflor on Bemisia tabaci Genn. (Hemiptera: Aleyrodidae): Insights into toxicity, biochemical impact, and molecular docking

Authors' Responses to “Reviewer 2” comments

Dear Professor,

We deeply appreciate your commitment to reviewing our manuscript and offering valuable feedback for its improvement. Your constructive comments have been carefully integrated into our revised version. Below, you'll find a summary of our responses to your suggestions.

Thank you once again for your invaluable contribution to the refinement of our work.

Kind regards,

Authors

Top of Form

The manuscript “The synergistic effect of lemongrass essential oil on flometoquin, flonicamid, and sulfoxaflor on Bemisia tabaci Genn. (Hemiptera: Aleyrodidae): Insights into toxicity, biochemical impact, and molecular docking” (insects-2967128) reported the synergistic effect of lemongrass essential oil on flometoquin, flonicamid, and sulfoxaflor on B. tabaci. The results of this study provide valuable information for mitigating the current status of insecticide resistance in B. tabaci.

****Authors Response

We thank the referee for his positive evaluation of the MS. All comments and suggestions are addressed below.

  1. The authors failed to provide a coherent rationale for the selection of lemongrass essential oil as a promising candidate in the Introduction section. It is recommended that the investigation into the utilization of lemongrass essential oil for pest control and managing insecticide resistance be incorporated.

****Authors Response

We followed the referee's suggestion and added a paragraph in the introduction section that emphasizes the importance of utilization of lemongrass essential oil for pest control and managing insecticide resistance. (Lines 79-83)

  1. The mode of action and target site of Flometoquin, a newly developed insecticide, necessitate detailed elucidation within scholarly discourse.

****Authors Response

We have incorporated the referee's suggestion by adding the mode of action and target site of Flometoquin on lines 89-90. We appreciate the valuable input and have ensured that this important information is now included in the manuscript. Thank you for your guidance.

  1. In the Materials and Methods section, it is imperative for the authors to furnish a comprehensive account of the protocol utilized to ascertain detoxification enzyme activity, encompassing key variables like incubation duration, temperature conditions, and pH levels. Such meticulous documentation serves to bolster the integrity of the findings, notably in substantiating the absence of noteworthy discrepancies in GST and P450 enzyme functionalities. The following reference might be helpful: Journal of Agricultural and Food Chemistry, 2023, 71, 5230−5239.

****Authors Response

Concerning to the biochemical analysis conditions, we used the same conditions that our lab used in the following references. The references have been added in the text to avoid the repetition.

  1. Moustafa, M.; Hefny, D.; Alfuhaid, N.; Helmy, R.; El-Said, N.; Ibrahim, E.-D. Effectiveness and Biochemical Impact of Flubendiamide and Flonicamid Insecticides against Bemisia Tabaci (Hemiptera: Aleyrodidae) and Residue Dissipation in Cherry Tomato Plants and Soil under Greenhouse Conditions 1. J. Entomol. Sci. 2024, 59, 1–22, doi:10.18474/JES23-61.
  2. Moustafa, M.; Awad, M.; Amer, A.; Hassan, N.; Ibrahim, E.-D.; Ali, H.; Akrami, M.; Salem, M. Insecticidal Activity of Lemongrass Essential Oil as an Eco-Friendly Agent against the Black Cutworm Agrotis Ipsilon (Lepidoptera: Noctuidae). Insects 2021, 12, 737, doi:10.3390/insects12080737.
  3. Moustafa, M.A.M.; Hassan, N.N.; Alfuhaid, N.A.; Amer, A.; Awad, M. Insights into the Toxicity, Biochemical Activity, and Molecular Docking of Cymbopogon Citratus Essential Oils and Citral on Spodoptera Littoralis (Lepidoptera: Noctuidae). J. Econ. Entomol. 2023, 116, 1185–1195, doi:10.1093/jee/toad093.

In addition, the suggested comments for cytochrome p450 has been added in the text.

Thus, we followed the referee's suggestion regarding GST determination and used the suggested citation (lines 204-205 & 207-208).

  1. Please confirm the presentation of Figure 1, ensuring uniformity in the measurement units of enzyme activities across both Figure 1 and Figure 2.

****Authors Response

Done.

  1. The reliability of the control group within the results of GST enzyme activity appears to be insufficient; it is advisable to carefully review the provided data.

****Authors Response

We apologize for the typographical error in the measurement unit. The correct unit for GST measurement should be nmol/min/mg protein instead of mmol/min/mg protein as mistakenly typed in Figures 1 and 2. This modification has been made accordingly.

Thank you once again for your attention to detail and for providing us with such precise feedback.

  1. Based on the discoveries outlined in sections 3.3 and 3.4, it is imperative to elucidate and deliberate on the diverse patterns noted in enzyme activities subsequent to insecticide treatments at LC25 and LC50 levels. Notably, the marked reduction in AChE activity post-LC25 flometoquin exposure, juxtaposed with the absence of notable variance in the context of LC50 treatment, demands additional scrutiny and discourse. Analogous patterns evident in alternative treatments similarly necessitate thorough contemplation.

****Authors Response

The toxicological results contrast with the acetylcholinesterase inhibitory activity of this insecticide, considering that AChE inhibition is one of the widely recognized mechanisms of its involvement in insecticidal activity. However, The use of lower concentrations of the insecticidal compound can lead to inhibition of acetylcholinesterase, leading to acetylcholine (AChE) accumulation and overstimulation of AChE receptors at synapses of the autonomic nervous system and neuromuscular junctions.

  1. The study employed the active sites of AChE in Drosophila melanogaster for analysis. Consideration is suggested for utilizing those in B. tabaci or conducting a comparative evaluation of the active sites between the two species. A more detailed account of the statistical analysis of docking results is recommended within the Material and Methods section.

****Authors Response

Comment 1: Active site consideration for B. tabaci and comparative evaluation with Drosophila melanogaster

Thank you for your comments regarding the choice of the active site for our molecular docking studies. In our study, we specifically chose the acetylcholine esterase (AChE) from Drosophila melanogaster (PDB ID: 6XYS) based on its relevance to our research focus and the availability of its crystal structure in the Protein Data Bank. The use of this particular enzyme is supported by its well-documented crystal structure, which provides a reliable foundation for our docking analysis. While comparative analysis with Bemisia tabaci AChE could be insightful, unfortunately, the crystal structures or reliable homology models of AChE from B. tabaci are not sufficiently documented or available in public repositories such as RCSB PDB, which limits their utility for precise molecular docking studies.

Furthermore, the aim of our study was not to compare the enzymatic activity across species but to assess the binding efficiency and mode of various compounds to a known AChE structure. As our objective was focused on the interaction dynamics within a specific enzyme that downregulated due to our tested compounds, the use of D. melanogaster’s AChE was deemed most appropriate for the goals of this study.

Comment 2: Detailed account of statistical analysis of docking results

Regarding the statistical analysis of our docking results, it is important to note that molecular docking studies, by their nature, are typically deterministic and do not always lend themselves to traditional statistical analysis such as p-values or confidence intervals which are common in other forms of quantitative research. Docking studies primarily rely on the computational prediction of the binding energies and the exploration of possible binding modes of ligands within the enzyme active sites. The results are generally assessed based on the docking scores and the stability of the ligand-protein interactions over simulation runs.

In our methodology, we validated the docking procedure by re-docking the co-crystallized ligand in its original enzyme structure to ensure the reproducibility and reliability of the docking positions and scores. The binding energies provided in our results (Table 3) serve as direct indicators of the binding affinity and are compared to known control compounds. This approach, although not statistical in a conventional sense, is a standard practice in molecular docking studies to ensure the accuracy and the predictive power of the docking simulations.

We appreciate the opportunity to clarify these points and thank you for your insightful suggestions which might be considered for future research directions.

  1. While the scoring of proteins and small molecules typically relies on the Affinity Bond strength parameter, the authors' criterion of the Energy score reveals diverging trends compared to the Affinity Bond strength.

****Authors Response

We appreciate the reviewer's attention to the details of our docking study, specifically regarding our choice of metrics for evaluating ligand-protein interactions. In our research, we have chosen to prioritize energy scores as the primary metric for assessing the efficacy of ligand binding to the acetylcholine esterase (AChE).

Energy scores, such as those obtained from our molecular docking analyses, provide a comprehensive reflection of the total interaction energy between a ligand and its target protein. This includes contributions from hydrogen bonding, hydrophobic effects, van der Waals interactions, and electrostatic forces. The energy score essentially offers an aggregate measure of the stability and favorability of the ligand-receptor complex formed, which is crucial for predicting the biological efficacy of potential inhibitors.

In contrast, affinity bond strength, while insightful, represents a more localized measure of interaction between specific pairs of atoms. While valuable, these measurements alone do not always accurately predict the overall binding affinity, as they do not account for the entire interaction landscape within the active site. Thus, focusing solely on these parameters might provide a misleading interpretation of a compound's effectiveness as a drug candidate.

Our results, as detailed in the results section and Table 3, demonstrate that the selected compounds exhibit higher overall energy scores compared to the co-crystallized ligand of the evaluated enzymes. This suggests that these compounds may form more stable complexes with the AChE enzyme, potentially translating into higher inhibitory activity. These findings are what the scientific community of computational drug design and discovery relies on.

Moreover, it is noteworthy that molecular docking techniques inherently integrate various interaction parameters to calculate the energy score. These techniques are widely accepted in the field for preliminary assessments of compound efficacy. Our study aligns with standard practices within the discipline and leverages these metrics to ensure the reliability and relevance of our findings.

We believe that our methodological approach, which emphasizes a holistic view of ligand binding through energy scores, is not only justified but also provides valuable insights that are critical for the further development of AChE inhibitor.

  1. The visual representation depicted in Figure 3 exhibits substandard quality as the binding sites remain imperceptible. An in-depth analysis centering on the binding site is imperative to enhance the scholarly discourse.

****Authors Response

We are grateful for your comments on the visual elements of our publication. We understand the importance of clear and informative figures in enhancing the comprehension and impact of our findings. Our figures were meticulously designed to provide an insightful and detailed depiction of the molecular interactions between the ligands and the acetylcholine esterase (AChE) enzyme, which is critical for understanding the mechanism of inhibition proposed in our study.

Specifically, the figures in question present both 2D and 3D interaction diagrams that elucidate the docking postures of the compounds within the active site of the AChE enzyme (PDB ID: 6XYS). The 2D diagrams effectively highlight the nature of the interaction – whether it involves hydrogen bonds, hydrophobic interactions, or arene contacts – thus providing a comprehensive view of the binding dynamics. The 3D representations are crafted to offer a spatial context to these interactions, displaying how the compounds fit within the active site contours and interact with key amino acid residues.

We have employed standard visualization techniques commonly accepted in the field, ensuring that our diagrams meet the scholarly standards for publication. These include the use of color coding to differentiate between types of interactions and residues, as well as the representation of hydrogen bonds and pi-interactions in distinct colors for clear differentiation. Our approach aligns with the best practices for depicting such complex biochemical information in a manner that is both scientifically accurate and accessible to the reader.

Adding to this we enhanced the resolution of the figure in question to exceed 600 PPI (pixels per inch) as it was set at 300 PPI, which is well above the standard resolution for high-quality printed material. This adjustment will ensure that all the intricate details of the molecular interactions are crisply represented and easily discernible.

Furthermore, these visualizations serve not merely as illustrative complements but as integral parts of our scientific argument, presenting a compelling visual narrative that corroborates our textual analysis. They allow us to convey complex data in an interpretable form, aiding in the validation of our docking results. In light of this, we firmly believe that our figures provide a clear, accurate, and valuable visual representation of our findings, contributing positively to the scholarly discourse. We have received constructive feedback on these visuals from experts in the field, further affirming their quality and effectiveness.

Round 2

Reviewer 2 Report

Comments and Suggestions for Authors

The authors have addressed all my comments.